# Large scale paired antibody language models

**Henry Kenlay**[1☯], **Frédéric A. Dreyer**[1☯]*, **Aleksandr Kovaltsuk**[1], **Dom Miketa**[1],
**Douglas Pires**[1], **Charlotte M. Deane**[1,2]

**1** Exscientia, Oxford Science Park, Oxford, United Kingdom, **2** Department of Statistics, University of Oxford, Oxford, United Kingdom

☯ These authors contributed equally to this work.
* dreyer.frederic@gene.com

## Abstract

Antibodies are proteins produced by the immune system that can identify and neutralise a wide variety of antigens with high specificity and affinity, and constitute the most successful class of biotherapeutics. With the advent of next-generation sequencing, billions of antibody sequences have been collected in recent years, though their application in the design of better therapeutics has been constrained by the sheer volume and complexity of the data. To address this challenge, we present IgBert and IgT5, the best performing antibody-specific language models developed to date which can consistently handle both paired and unpaired variable region sequences as input. These models are trained comprehensively using the more than two billion unpaired sequences and two million paired sequences of light and heavy chains present in the Observed Antibody Space dataset. We show that our models outperform existing antibody and protein language models on a diverse range of design and regression tasks relevant to antibody engineering. This advancement marks a significant leap forward in leveraging machine learning, large scale data sets and high-performance computing for enhancing antibody design for therapeutic development.

**Data Availability Statement:** The original studies and associated datasets for all sequence data used to train the model can be found at https://opig.stats.ox.ac.uk/webapps/oas/, with the list of the unpaired data accessible at https://opig.stats.ox.ac.

## Author summary

This study introduces IgBert and IgT5, two advanced antibody-specific language models capable of processing the extensive data generated by next-generation sequencing of antibodies. These models are trained on over two billion unpaired and two million paired antibody sequences from the Observed Antibody Space dataset. IgBert and IgT5 demonstrate state-of-the-art performance compared to existing models in tasks such as sequence recovery, expression, and binding affinity predictions. By employing comprehensive machine learning techniques and leveraging large-scale datasets, these models significantly enhance the potential for antibody design and therapeutic development. The trained models are publicly available, providing valuable tools for ongoing research in antibody engineering.

uk/webapps/oas/oas_unpaired/, and the list of paired data accessible at https://opig.stats.ox.ac.uk/webapps/oas/oas_paired/. To reach the original studies and associated datasets, users need to click the "search" button with no filters. The models weights are available on Zenodo at https://doi.org/10.5281/zenodo.10876909. The code used is available through the HuggingFace library at https://huggingface.co/exscientia.

**Funding:** The author(s) received no specific funding for this work.

**Competing interests:** The authors have declared that no competing interests exist.

## Introduction

Antibodies are proteins that play a central role in the adaptive immune system recognising and neutralising pathogens. They are symmetric Y-shaped molecules consisting of two light and two heavy chains (as depicted in Fig 1), and can identify and bind with high specificity to antigens—substances that the body identifies as foreign, such as bacteria, viruses, or toxins. Antibodies possess a variable region at their tip that allows them to recognise antigens, which is one of the ways they activate further immune functions. An antibody's specificity is primarily influenced by the complementarity-determining regions (CDRs) of the variable domain. The third CDR loop of the heavy chain is the most structurally diverse and typically the most important for antigen recognition [1–3]. The ability to specifically target pathogens makes antibodies a central element in both natural immune responses and medical applications.

The emergence of next-generation sequencing (NGS) technology has transformed our understanding of the diversity and function of antibodies [4]. NGS enables the rapid sequencing of antibody repertoires, revealing a detailed insight into the natural diversity of the variable region of antibodies. This extensive sequencing data provides a large-scale representation of the various antibodies that natural systems generate, and contains crucial information about the numerous ways in which antibodies can adapt to recognise a wide array of antigens. This data is key to understanding how the immune system responds to infections and how it can be manipulated for therapeutic purposes.

The vast amount of data generated by antibody sequencing presents a unique challenge, which traditional bioinformatics methods such as sequence alignment and motif search can only partially address. Protein language models, inspired by advances in natural language processing, offer an orthogonal approach, which have found a wide range of uses in protein design by learning rich representations from large corpora of unlabelled sequences. They can be trained specifically to learn the complex grammar of antibody sequences, often through a masked language model objective, where a random fraction of amino acid residues are hidden and the model is then tasked to reconstruct the original sequence. This unsupervised pre-training allows the model to learn a high dimensional contextualised embedding space on which sequences can be mapped, enabling a wide range of applications such as restoring missing residues [5], *de novo* sequence generation [6, 7], structure [8, 9] and property prediction such as binding sites [10], humanness [11] or thermostability [12]. Language models thus represent a

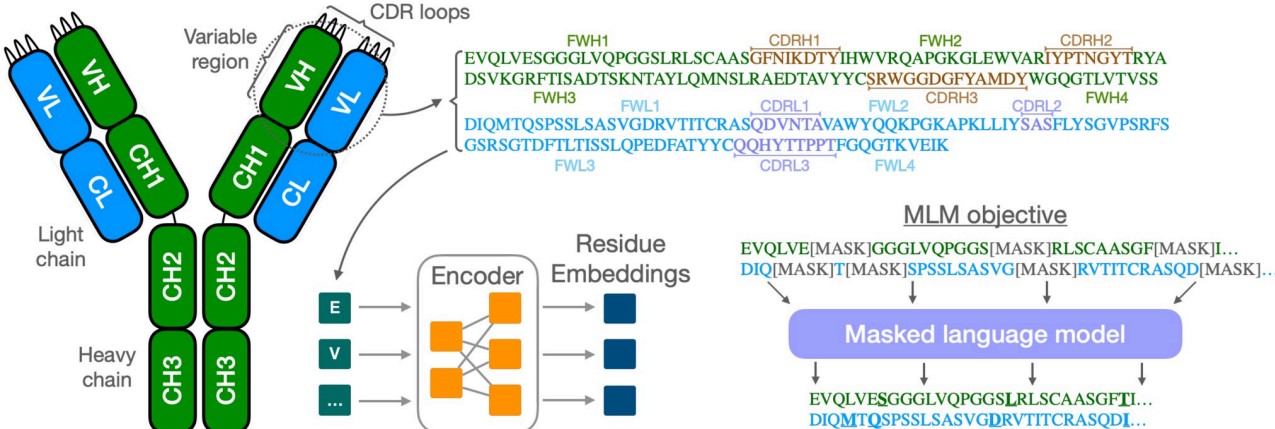

**Fig 1. Overview of an antibody structure and its domains.** The sequence of the variable region is used as input to the transformer encoder to obtain a residue-level embedding representation. Training is achieved through masked language modelling, where a random fraction of the input is replaced by mask tokens.

crucial tool in translating the vast antibody sequencing data into insights that can be used to advance therapeutic developments.

In this work, we train antibody language models on the Observed Antibody Space (OAS) [13, 14]. We consider both a BERT [15] and a T5 [16] model, which are initialised with weights of a general protein model and pre-trained on a large corpus of unpaired antibody sequences, consisting of the totality of known unpaired sequences for the BERT model, and 700M cluster representatives for the T5 model. These models are then fine-tuned on a smaller dataset of all known paired heavy and light variable region sequences. To our knowledge, these are the largest and one of the only paired antibody language models, as well as the first T5 antibody language model trained for sequence encoding to date. We then show how our models perform on downstream applications, notably sequence recovery, as well as expression and binding affinity predictions, and demonstrate that they outperform existing state-of-the-art protein and antibody language models on these key tasks.

The models trained as part of this study are made publicly available and can be readily used in a range of tasks of relevance to antibody engineering [17], as detailed in S1 Appendix.

## Related work

The vast amount of sequence data available in public repositories positions unsupervised representation learning approaches as powerful ways to leverage machine learning in biology. Protein language models, notably transformer-based architectures such as Evolutionary Scale Modeling (ESM) [18, 19], ProtTrans [20], and ProGen [21] have been pivotal in protein structure prediction and function annotation, providing a solid foundation for generalised protein modeling. Trained on extensive evolutionary data containing billions of sequences [22, 23], these models have also demonstrated remarkable capabilities in understanding protein sequences, their evolutionary trajectories and constraints [24].

Turning to antibody-specific literature, BERT-like models [15, 25] such as AbLang [5], AntiBERTy [26], Sapiens [11], and AntiBERTa [10] have shown that training language models on sequences from the immunoglobulin protein superfamily can be of particular use, notably to restore missing residues, to study affinity maturation trajectories and to identify paratope residues (*i.e.*, those responsible for recognising antigen binding). Models trained with paired sequences of heavy and light chains can learn cross-chain features, which has been shown to improve performance on predictive downstream tasks [27, 28]. Effective antibody-specific models can also be obtained by fine-tuning general protein models on antibody sequences [27, 29], which can often be less computationally intensive than training a model from random initial weights.

While most antibody language models have focused on encoder-only architectures, some generative models have explored encoder-decoder [30] and decoder-only models [6, 31].

## Methods

### Models

In this study, we focus on two language model architectures, BERT [15] and T5 [16]. BERT is an encoder-only model focused on generating contextual embeddings for each token in an input sequence whereas T5 is an encoder-decoder model designed for sequence-to-sequence tasks, where the encoder processes the input sequence and the decoder generates the output sequence. Both are based on transformer architectures [32], where inputs are processed through a series of blocks that incorporate self-attention with feed-forward connections. The self-attention component allows the network to construct representations that incorporate

context from anywhere in the sequence, and which can directly encode pairwise interactions between amino acid residues.

Bidirectional Encoder Representations from Transformers (BERT) is a bidirectional model which learns a contextual representation by predicting randomly masked tokens in the input. By incorporating information from both left and right context across each layer, it can capture nuanced understanding of language structure and context and has seen diverse applications in natural language processing. It is typically pre-trained on a large corpus of text, which in our case are protein and antibody sequences, and then fine-tuned for specific tasks, including but not limited to sentiment analysis, language translation, or in the case of proteins, structural and functional property prediction.

Text-To-Text Transfer Transformer (T5) frames all tasks as text-to-text problems, and employs span-based masking where contiguous random spans are replaced with sentinel tokens. The decoder uses a causal attention mechanism, which allows it to handle a diverse range of tasks including translation, summarisation and question answering with a unified approach.

We train both models with a masked language modelling (MLM) objective. Sequences are parsed through a tokeniser which maps each amino acid type in the sequence to a distinct token, and includes a separator token at the end of each chain. The BERT tokeniser has a classification and mask token, while the T5 tokeniser has a single sentinel token which is used as an equivalent to a mask token. During training, each input sequence is corrupted by replacing a fraction of the amino acid residues with a special mask token for the BERT model, while for the T5 model we follow [20], with a corresponding single-token span. The models are then trained to predict these tokens from the corrupted sequence, such that for a sequence with $n$ residues $\mathbf{s} = (s_1, \ldots, s_n)$ the loss is defined as

$$\mathcal{L}_{\text{MLM}} = \mathbb{E}\left[\sum_{i \in \mathcal{M}} - \log p(s_i | \mathbf{s}_{/\mathcal{M}})\right], \tag{1}$$

where $\mathcal{M}$ are all the masked positions in a sequence and $\mathbf{s}_{/\mathcal{M}}$ denotes the masked sequence where all masked positions have been replaced by a mask token.

A key difference between the BERT and T5 models on this MLM task is that the BERT model has a simple MLM head to predict the token based on the encoded embedding, while the T5 model is trained to predict the sequence as a text-to-text task and relies on both the encoder and decoder, even if only the encoder is later used in most downstream tasks. The loss is then computed over the full sequence for the T5 model, while only the masked residues contribute to the BERT loss.

For both the T5 and BERT model, we train our models starting from existing weights of a general protein language model that was pre-trained on protein sequences from the UniRef [33] and BFD [34] databases. We use as starting point the best performing BERT-style model, ProtBert, and the state-of-the-art ProtT5 model, both from the ProtTrans family of protein transformer models [20]. We found that this approach generally achieves higher performance for a fixed compute budget than training from random weights, as was similarly reported in [27].

## Data preparation

We train our models on sequences from OAS, a database of predominantly human immune repertoires which contains over two billion unpaired variable heavy and light chain sequences, as well as over two million paired variable region sequences. For a brief summary of the OAS dataset, see S2 Appendix. An overview of our training strategy is shown in Fig 2, which

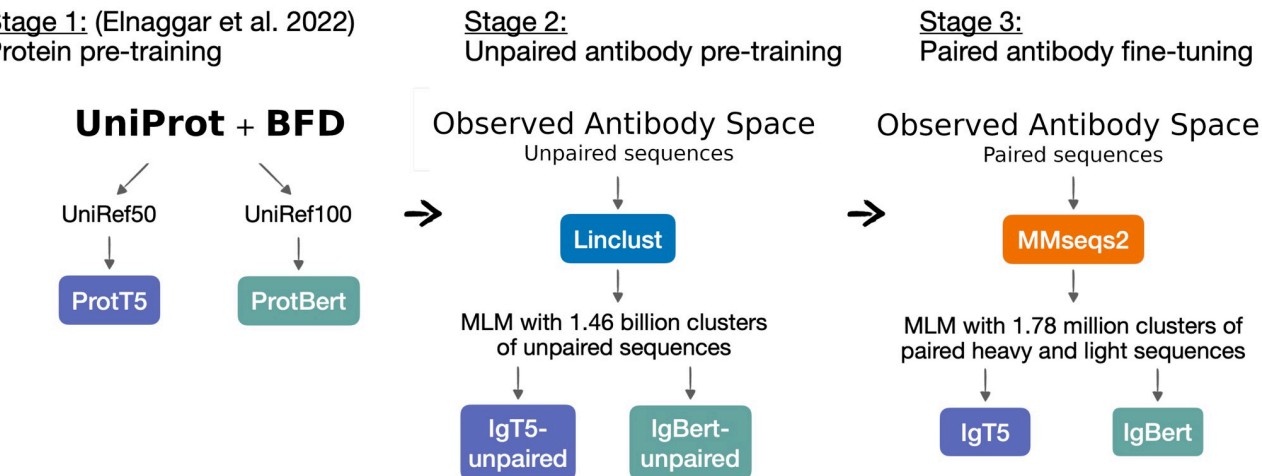

**Fig 2. Data processing and training strategy.** We further pre-train the ProtT5 and ProtBert models from [20] on unpaired antibody sequences from OAS after clustering them with Linclust. These unpaired models are then fine-tuned on paired sequences clustered with MMseqs2, combining the two variable region chains into a single input with a separator token.

includes the initial pre-training on general protein sequences from UniRef [33] and BFD [34] and initial weights from the ProtTrans models [20].

We perform a second stage pre-training on the unpaired OAS sequences. We take all 2,097,593,973 unique sequences from the database after removing duplicates. We cluster the sequences to ensure no sequences in the validation and test sets are near-duplicates to those in the training set. Due to the large size of this dataset, we cluster these sequences with Linclust [35], an approximate clustering algorithm that scales linearly with the number of sequences. We use 95% sequence identity as a threshold to create a dataset of 1,456,516,479 clusters. From these clusters, 40M are randomly selected to construct test and validation clusters. The remaining 1,416,516,479 clusters compose the training dataset, which contains 2,040,122,161 unique sequences. The validation and test datasets consist of 20M representative sequences each, selected from the 40M clusters selected above.

The last stage is fine-tuning on paired sequences. Here we consider the 2,038,528 unique paired antibody variable region sequences from OAS, and cluster them with MMseqs2 [36] using a 95% sequence identity threshold on the concatenated heavy and light chains. In contrast to Linclust, MMseqs2 is an exact clustering algorithm which ensures non-overlapping clusters, but scales quadratically with the number of sequences. This provides us with a dataset of 1,777,722 clusters, of which 40,000 are randomly selected as test and validation clusters. The training dataset consists of 1,992,786 unique sequences, distributed among 1,737,722 clusters. As validation and test data, we use the 20,000 representative sequences drawn from the test and validation clusters.

## Pre-training on unpaired sequences

All models were trained using 32 A100 GPUs using a distributed DeepSpeed ZeRO Stage 2 strategy, which shards optimiser states and gradients for memory efficiency [37]. We fix the number of training steps ahead of time and linearly warm-up the learning rate to the maximum learning rate before linearly annealing to zero. The optimiser for both models is AdamW with a weight decay parameter of $10^{-5}$ for regularisation [38]. Each sequence was padded to a length of 200. We summarise the model and training parameters for all models in Table 1.

**Table 1. Hyperparameters for all models trained in this study.** The IgBert-unpaired and IgT5-unpaired are initialised with the ProtBert and ProtT5 weights respectively. The IgBert model is trained starting from the IgBert-unpaired weights, and the IgT5 model is derived from IgT5-unpaired.

| Parameters | IgBert-unpaired | IgT5-unpaired | IgBert | IgT5 |
|---|---|---|---|---|
| Initial weights | ProtBert | ProtT5 | IgBert-unpaired | IgT5-unpaired |
| Number of layers | 30 | 24 | 30 | 24 |
| Size hidden layers | 1024 | 1024 | 1024 | 1024 |
| Intermediate size hidden layers | 4096 | 16,384 | 4096 | 16,384 |
| Number of attention heads | 16 | 32 | 16 | 32 |
| Dropout | 0.0 | 0.1 | 0.0 | 0.1 |
| Target length | 200 | 200 | 300 | 300 |
| Batch size per GPU | 300 | 24 | 192 | 15 |
| Global batch size | 9600 | 768 | 6144 | 480 |
| Optimiser | AdamW | AdamW | AdamW | AdamW |
| Max learning rate | $10^{-5}$ | $5 \cdot 10^{-5}$ | $10^{-5}$ | $5 \cdot 10^{-5}$ |
| Weight decay | $10^{-5}$ | $10^{-5}$ | $10^{-5}$ | $10^{-5}$ |
| Warm-up steps | 10,000 | 30,000 | 2700 | 4000 |
| Total training steps | 212,512 | 911,500 | 45,000 | 60,000 |
| Masking probability | 0.15 | 0.15 | 0.15 | 0.15 |
| Number of parameters | 420M | 3B | 420M | 3B |

**IgBert-unpaired.** The IgBert-unpaired model is initialised with ProtBert weights. We fine-tune on all unpaired OAS sequences that belong to training clusters. The batch size is set to 300 per GPU for a global batch size of 9600.

To complete a full epoch we trained for 212,512 steps. We used a linear annealing scheduler with a max learning rate of $10^{-5}$ and 10,000 warm-up steps. Training time was approximately 66 hours.

We used a MLM objective where 15% of tokens were selected for masking. Of these 80% are replaced by a masked token, 10% are changed to a random token in the vocabulary and 10% are left unchanged.

**IgT5-unpaired.** The T5 architecture is much larger than the BERT architecture in terms of parameters leading to a larger memory and compute footprint. As such, we train on just representative sequences of the training clusters for half an epoch due to compute constraints. Similar to before, we initialise training using the ProtT5 weights.

The batch size is 24 per GPU for a global batch size of 768. We trained for a total of 911,500 steps, which corresponds to 700M representative sequences or half of the unpaired cluster representatives. The max learning rate was set to a larger $5 \times 10^{-5}$, in line with the proportionally larger learning rate used to train ProtT5 compared to ProtBert [20]. The warm-up was 30,000 steps. Training time was approximately 9 days.

Following ProtT5, we used the T5 objective with a maximum span length of one. This leads to a similar loss to the MLM objective, but with an additional sequence reconstruction term. We masked 15% of tokens where 90% are replaced by a mask token and 10% are replaced by a random amino acid token.

## Fine-tuning on paired sequences

We further tune our unpaired models on paired sequences available in the OAS. To prevent catastrophic forgetting [39, 40], we use batches with both unpaired and paired data, incorporating two unpaired sequences into the batch for each paired sequence included. Paired

sequence inputs are constructed by concatenating the heavy and light chain sequences, inserting a separator token in between them.

The input sequences were padded to 300 to account for the increased length of paired sequences, reducing the batch size we could fit into memory for both models. Apart from the learning rate schedule and batch construction the rest of the setup remained unchanged. We fine-tuned both models for 16 hours.

**IgBert.**   IgBert was trained by fine-tuning IgBert-unpaired using a batch size of 128 unpaired and 64 paired sequences per GPU. The paired training data consists of all sequences in the training clusters for the paired sequences, and randomly chosen unpaired sequences from the unpaired training clusters to complete the batches. We trained for approximately 46 epochs (45,000 steps), defining an epoch as one pass over the paired OAS dataset. The warm-up consisted of 2,700 steps, approximately three epochs. We use the same MLM objective as in the unpaired pre-training, where 15% of the input is masked, 80% of those are replaced by a mask token, 10% by a random token and 10% are kept identical.

**IgT5.**   Similar to IgBert, we used IgT5-unpaired as our starting weights to train IgT5. The batch size was 10 unpaired and 5 paired sequences per GPU. The batches are constructed from all sequences from the paired training clusters, and representative sequences randomly selected from the unpaired training clusters which have not been seen in the unpaired pre-training. We trained for approximately 5 epochs (60,000 steps) of the paired OAS data, with 4,000 warm-up steps. The objective is identical to the unpaired pre-training case, with 15% masked tokens, from which 90% are replaced by a sentinel token and 10% are randomly replaced by a different amino acid.

## Results

We evaluate our models on a range of downstream tasks, comparing with existing antibody and protein language models. As baselines, we consider the original ProtTrans protein language models [20], as well as the open-source antibody language models AbLang [5] and AntiBERTy [26]. We examine two tasks of interest: the sequence recovery for each component of the antibody variable region, and performance in predicting experimental binding affinity or expression data from language model embeddings.

### Sequence recovery

In Table 2, we show the sequence recovery achieved after masking 15% of residues in paired sequences from the test dataset at random. Similar results for unpaired sequences are provided in S3 Appendix. The recovery fraction is shown separately for each framework and CDR loop, as well as over the total heavy and light sequences.

This task benefits from training on antibody-specific data, as can be seen by the comparatively poor performance of the protein language models, notably in the hypervariable H3 and L3 loops. We also observe that the IgT5 model achieves the highest recovery across all but one of the CDR loops, CDRL1, where IgBert achieves marginally higher accuracy. For sequence recovery in the framework regions, both IgBert and IgT5 have comparable high performance.

### Binding affinity and expression

We now consider the language model embeddings as input feature to a predictive downstream task. To this end, we consider the largest datasets from the FLab benchmark [41], which include the binding energy datasets from [42] with 422 data points, from [43] with 2048 data points and from [44] with 4275 data points, as well as the corresponding expression data from this last study.

**Table 2. Fraction of correctly predicted residues by region (frameworks (FW) and CDRs for the heavy (H) and light (L) chains), after masking 15% of the sequence for a test set of 20k paired heavy and light sequences.** The best, second and third best performing models for each region are shown in bold, underlined and italic respectively.

| Model | FWH1 | FWH2 | FWH3 | FWH4 | CDRH1 | CDRH2 | CDRH3 | Total VH |
|---|---|---|---|---|---|---|---|---|
| AbLang [5] | *0.979* | 0.967 | *0.956* | 0.981 | *0.910* | *0.884* | 0.593 | 0.910 |
| AntiBERTy [26] | 0.978 | 0.965 | 0.954 | 0.977 | 0.907 | 0.882 | 0.521 | 0.900 |
| ProtBert [20] | 0.802 | 0.761 | 0.738 | 0.846 | 0.656 | 0.456 | 0.277 | 0.682 |
| IgBert-unpaired | 0.979 | 0.965 | 0.955 | 0.980 | 0.904 | 0.884 | 0.592 | 0.910 |
| IgBert | <u>0.981</u> | **0.969** | **0.958** | *0.981* | <u>0.913</u> | <u>0.886</u> | *0.601* | <u>0.913</u> |
| ProtT5 [20] | 0.904 | 0.854 | 0.888 | 0.914 | 0.753 | 0.629 | 0.339 | 0.793 |
| IgT5-unpaired | 0.979 | *0.967* | *0.956* | <u>0.982</u> | 0.909 | 0.884 | <u>0.603</u> | *0.912* |
| IgT5 | **0.982** | <u>0.969</u> | <u>0.957</u> | **0.983** | **0.915** | **0.894** | **0.620** | **0.916** |
| Model | FWL1 | FWL2 | FWL3 | FWL4 | CDRL1 | CDRL2 | CDRL3 | Total VL |
| AbLang [5] | 0.966 | 0.968 | 0.971 | 0.962 | 0.891 | 0.901 | 0.838 | 0.949 |
| AntiBERTy [26] | 0.979 | *0.969* | 0.975 | 0.966 | 0.907 | 0.895 | 0.844 | 0.955 |
| ProtBert [20] | 0.660 | 0.786 | 0.783 | 0.634 | 0.469 | 0.438 | 0.290 | 0.665 |
| IgBert-unpaired | 0.980 | 0.970 | 0.974 | *0.966* | *0.908* | 0.898 | *0.846* | *0.956* |
| IgBert | **0.988** | **0.974** | <u>0.981</u> | <u>0.974</u> | **0.923** | <u>0.915</u> | <u>0.863</u> | <u>0.965</u> |
| ProtT5 [20] | 0.846 | 0.901 | 0.880 | 0.850 | 0.696 | 0.604 | 0.517 | 0.820 |
| IgT5-unpaired | *0.981* | 0.967 | *0.975* | 0.917 | 0.908 | *0.909* | 0.842 | 0.951 |
| IgT5 | <u>0.988</u> | <u>0.973</u> | **0.981** | **0.978** | <u>0.922</u> | **0.916** | **0.869** | **0.966** |

For each dataset, we use a linear least squares fit with $L_2$ regularisation on the embedding representation features assessed under 10-fold cross-validation. We use a 5-fold inner cross-validation to select the regularisation hyper-parameter $\lambda \in \{1, 10^{-1}, \ldots, 10^{-6}, 0\}$. For the paired IgT5 and IgBert models, the variable region embedding is computed as the average over both the heavy and light chain tokens, while for all other models, we compute separately an embedding representation for the heavy and light chain by computing the average over their token representations, and then concatenate the heavy and light sequence vectors into a higher-dimensional feature vector. Thus, the paired models require cross-chain features that persist after averaging, and are condensed to a feature vector of half the size of the unpaired models.

Despite this, as can be seen in Table 3, the highest $R^2$ on the binding energy prediction tasks comes from the paired models, demonstrating the importance of training on natively paired

**Table 3. $R^2$ for a linear model applied on the embeddings of each language model to predict binding energy to an antigen or expression.** The best, second and third best performing models for each benchmark are shown in bold, underlined and italic respectively.

| Model | Binding | Binding | Binding | Expression |
|---|---|---|---|---|
| | $N = 422$ | $N = 2048$ | $N = 4275$ | $N = 4275$ |
| | [42] | [43] | [44] | [44] |
| AbLang [5] | *0.293 ± 0.117* | <u>0.246 ± 0.038</u> | <u>0.244 ± 0.034</u> | 0.439 ± 0.027 |
| AntiBERTy [26] | 0.239 ± 0.102 | 0.217 ± 0.056 | 0.199 ± 0.025 | 0.401 ± 0.032 |
| ProtBert [20] | 0.200 ± 0.106 | 0.149 ± 0.024 | 0.101 ± 0.017 | 0.491 ± 0.029 |
| IgBert-unpaired | 0.278 ± 0.094 | 0.181 ± 0.040 | 0.177 ± 0.018 | 0.347 ± 0.023 |
| IgBert | **0.306 ± 0.114** | 0.131 ± 0.047 | 0.174 ± 0.032 | 0.400 ± 0.023 |
| ProtT5 [20] | 0.290 ± 0.105 | 0.186 ± 0.037 | *0.206 ± 0.029* | **0.697 ± 0.02** |
| IgT5-unpaired | <u>0.299 ± 0.119</u> | *0.245 ± 0.049* | 0.179 ± 0.014 | <u>0.567 ± 0.025</u> |
| IgT5 | 0.274 ± 0.070 | **0.297 ± 0.057** | **0.25 ± 0.019** | *0.548 ± 0.067* |

data, which was also recently highlighted by [27]. The general protein models, ProtBert and ProtT5, perform below the antibody-specific ones on binding prediction tasks. However, on the expression task, the general protein language models outperform the antibody specific models, with ProtT5 having the highest $R^2$. This suggests that evolutionary information or broader patterns across different protein families present in general protein models are important in general property downstream tasks and are not learned as effectively by antibody language models.

We use the $R^2$ metric due to its robustness in evaluating the predictive accuracy of our model across diverse datasets. A similar analysis showing the Pearson correlation can be found in S4 Appendix.

## Discussion

In this study, we investigated whether large antibody-specific language models could improve performance on sequence recovery and downstream predictive tasks.

We pre-trained both T5 and BERT models on over two billion unpaired sequences from the Observed Antibody Space using an MLM objective. We then fine-tuned the unpaired models on paired heavy and light sequences to create paired models that can learn cross-chain features. The resulting IgBert and IgT5 models are made available through Zenodo [17] and Huggingface. Models available at huggingface.co/Exscientia.

We identify several key takeaways that shed light on the potential and limitations of these specialised language models.

One striking result was the substantial improvement in performance achieved through fine-tuning our models on paired data. This improvement can be attributed to the models' ability to learn cross-chain features, facilitating a deeper understanding of antibody sequences. Furthermore, we believe that the quality of the paired data played a pivotal role in enhancing performance, highlighting the significance of data quality in training protein language models. Experimental verification of designs derived from IgT5 and IgBert would provide an interesting avenue for validating this performance. It would also be valuable to study the scaling of model performance with the amount of paired sequence data used in training, and whether additional sequencing data is likely to lead to substantial improvements in down stream tasks.

While our experiments demonstrated improved binding affinity predictions using embeddings from large antibody-specific language models, we also observed a discrepancy in their ability to predict expression compared to general protein models. This result suggests that general protein models, which have acquired knowledge about the evolutionary diversity of proteins, have an advantage in capturing patterns relating to protein expression, whilst antibody language models may be better suited at learning specialised features relevant to antibody specific properties.

The results presented in this article open the avenue for several promising future directions. A second stage pre-training approach that incorporates general protein sequences into the training data would allow the model to enhance generalisability, improving its performance on expression benchmarks. Additionally, the integration of structural and functional information into the language model embedding representations through the application of methods such as contrastive learning [45] might improve performance on property prediction tasks, and there has been promising recent work in this direction [46, 47]. It would also be interesting to consider more parameter efficient fine-tuning strategies such as low rank adaptation [48], and training strategies that can maintain expertise in previous tasks such as elastic weight consolidation [49] to enable more scalable and robust specialised protein language models. More extended ablation studies of the impact on model performance of

clustering thresholds used in preparing the training and test data would be valuable, as these have been shown to impact training dynamics in general protein language models [50]. Notably, it would be useful to understand whether different clustering choices, e.g. clustering along the CDRs instead of the variable region sequence, or training on cluster representatives instead of all sequences, impact training dynamics at scale for different sequence identity thresholds. The models trained in this study might provide a particularly powerful approach when exploring more intricate downstream tasks, such as structure prediction. Furthermore, the paired models offer an interesting opportunity to leverage learned cross-chain features to predict pairing relationships. These models could also be leveraged for *in silico* affinity maturation, incorporating them into a therapeutic candidate generation pipelines to improve developability [51, 52].

Extending the capabilities of our IgT5 model by fine-tuning it for generative tasks also opens up opportunities for generating novel antibody sequences that can provide a powerful approach for antibody design.

Our research on antibody-specific language models highlights their potential for advancing antibody-related research and development. The work presented in this article provides a key step towards harnessing the power of these models for therapeutic development, and paves the way for breakthroughs on the application of specialised language models for antibody engineering and drug discovery.

## Supporting information

**S1 Appendix. Model usage.** Example code for the use of the IgT5(-unpaired) and IgBert (-unpaired) models, along with links to more detailed information.
(PDF)

**S2 Appendix. OAS dataset.** Overview of the training dataset, which consists of the OAS database accessed in November 2023.
(PDF)

**S3 Appendix. Unpaired sequence recovery.** The sequence recovery achieved by each model on a test set of unpaired sequences.
(PDF)

**S4 Appendix. Correlation for binding affinity and expression.** The correlation of the predicted binding energy and expression with the data.
(PDF)

## Acknowledgments

We are grateful to the NVIDIA Corporation for providing access to the 4 DGX nodes used in this study. We thank Daniel Cutting and Constantin Schneider for useful discussions.

## Author Contributions

**Conceptualization:** Henry Kenlay, Frédéric A. Dreyer, Charlotte M. Deane.

**Data curation:** Frédéric A. Dreyer, Aleksandr Kovaltsuk.

**Formal analysis:** Henry Kenlay.

**Investigation:** Henry Kenlay, Frédéric A. Dreyer.

**Methodology:** Henry Kenlay, Frédéric A. Dreyer.

**Software:** Henry Kenlay, Frédéric A. Dreyer, Dom Miketa.

**Supervision:** Douglas Pires, Charlotte M. Deane.

**Validation:** Henry Kenlay, Frédéric A. Dreyer.

**Visualization:** Frédéric A. Dreyer.

**Writing – original draft:** Frédéric A. Dreyer.

**Writing – review & editing:** Henry Kenlay, Frédéric A. Dreyer, Aleksandr Kovaltsuk, Dom Miketa, Douglas Pires, Charlotte M. Deane.

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
