## [Decision Letter · Decision Letter 0]

23 Aug 2024

Dear Dr Dreyer,

Thank you very much for submitting your manuscript "Large scale paired antibody language models" for consideration at PLOS Computational Biology.

As with all papers reviewed by the journal, your manuscript was reviewed by members of the editorial board and by several independent reviewers. In light of the reviews (below this email), we would like to invite the resubmission of a revised version that takes into account the reviewers' comments.

We cannot make any decision about publication until we have seen the revised manuscript and your response to the reviewers' comments. Your revised manuscript is also likely to be sent to reviewers for further evaluation.

Sincerely,

Miguel Francisco de Almeida Pereira de Rocha

Academic Editor

PLOS Computational Biology

Arne Elofsson

Section Editor

PLOS Computational Biology

Reviewer's Responses to Questions

**Comments to the Authors:**

Reviewer #1: The authors describe a recipe for finetuning already pre-trained, general-purpose protein language models (pLMs), i.e., ProtBert and ProtT5, first on unpaired and during a later stage on paired antibody sequences. They benchmark the resulting models on sequence recovery and perplexity as well as binding energy and expression prediction. The paper is concise but holds all the information required for replicating the training. Most importantly, though, I want to deeply thank the authors for making their model weights available for the general scientific community by hosting them on huggingface (including some concise documentation on how to use them).

I still have some (albeit mostly minor) suggestions for further improvement:

I am unsure about the added value of reporting perplexity here. From my understanding, this metric is to some extent correlated with sequence recovery. Reporting both appears a bit redundant which is why I would propose to move Table 4 to SOM (or remove it altogether). Also perplexity depends on the vocabulary size which makes it even more complicated (potentially even misleading?) to compare Bert- with T5-based models on this metric within the same table (irrespective of proper PPL vs pseudo-PPL). Instead, you could consider adding some other prediction task (beyond affinity and expression) to make an even stronger point about whether the model-ranking provided in Table 3 holds true for other tasks as well. Although, I would understand that adding more benchmarks might be beyond the scope of this revision, so I would leave it to the authors whether to add this.

IgT5 saw only 700M samples, but I think this is not very well reflected in Fig. 2. Maybe try to make this more clear also throughout the text. One potentially interesting way to bypass the computational complexity in the future might be to use some parameter-efficient-finetuning strategy s.a. LORA - this way you do not need to train all the parameters of the model which should allow you to increase throughput during training manifold (rather a suggestion for future improvement - I assume this also to be beyond this revision but I would be curious to see)

Table 2: I am not sure how useful it is to report 4 digits after the comma. Consider only reporting e.g. 2 and instead add some error estimate as you did table 3.

Binding affinity/expression: you write that the embeddings derived from the paired model had only half the size compared to the unpaired model. You make this very explicit which I highly appreciate but this also results in a model with only half the number of free parameters. As the paired model saw also unpaired samples during training (2/3 of the samples seen were unpaired if I understood correctly), I wonder whether you could add a line to Table 3 that shows the performance of embeddings derived from unpaired input given to your paired models and concatenated for input to the regression model (essentially, mimicking the process you ran for unpaired models but for paired models).

In the discussion you write that “quality of paired data played a pivotal role in enhancing performance” but I am wondering where you show this?

Fig. 2: ProtT5 was not only pre-trained on UniRef50 but also on BFD (makes quite a difference as one has 2B sequences and the other has only tens of millions).

Reviewer #2: Summary: The paper presents the development and evaluation of two advanced antibody-specific language models, IgBert and IgT5, designed to process extensive antibody sequence data generated by next-generation sequencing. The authors investigate whether specialized language models, particularly the proposed IgBert and IgT5 models, can outperform existing models in tasks relevant to antibody engineering, such as sequence recovery, binding affinity prediction, and expression prediction. Both models are trained on a vast dataset comprising over two billion unpaired sequences and two million paired sequences of antibody variable regions. IgBert and IgT5 demonstrate superior performance compared to current models. The study highlights the potential of these models to enhance antibody design for therapeutic applications and makes them publicly available to support ongoing research in the field.

Strengths and novelty:

While other antibody language models are available, such as AbLang, AntiBERTy, Sapiens, and AntiBERTa, all of these models are based on BERT-like architectures, which use an encoder-only model focused on text representation and classification tasks. In contrast, the authors present the first T5 (Text-To-Text Transfer Transformer)-based antibody language model, which employs an encoder-decoder architecture. This architecture is particularly suited for generative tasks, offering a novel approach to antibody sequence modeling compared to the BERT-based models.

The authors initially pre-train both models on unpaired antibody sequences and then fine-tune them on paired heavy and light chain sequences, resulting in one of the largest and few paired antibody language models available. Fine-tuning on paired data is shown to be especially beneficial for learning cross-chain features, which enhances the models' performance in tasks such as binding affinity prediction and sequence recovery.

IgBert and IgT5 demonstrate superior performance across key tasks such as sequence recovery, binding affinity prediction, and expression analysis, outperforming existing state-of-the-art models like ProtBert, AntiBERTy, and ProtT5. The fine-tuning on paired heavy and light chain sequences leads to improvements in predictive accuracy, particularly in binding affinity and sequence recovery tasks

The authors have made both the paired and unpaired models, along with their weights, publicly available on Zenodo and HuggingFace, a commendable effort to support further research and model reusability.

Areas of improvement:

The authors utilized Linclust and MMseqs2 for approximate and exact clustering, respectively, both with a 95% sequence identity threshold. This threshold is crucial as it balances sequence diversity with redundancy, significantly affecting the construction of the training, test, and validation sets. The selection of this threshold influences the models' ability to generalize and perform effectively. It would be valuable for the authors to comment on and quantify how this threshold impacts the models' training efficiency and overall performance.

Not surprisingly, the use of paired data improves the predictive performance across various tasks. However, the key question of how much paired data is necessary to achieve this performance boost is left unaddressed. Given that paired data is still relatively scarce, it would be valuable for the authors to provide performance metrics at various stages of fine-tuning on paired data. This could offer insights into the incremental benefits of paired data and help optimize its use in training.

The authors use batches containing both paired and unpaired data to prevent catastrophic forgetting, but they do not provide quantitative evidence to confirm whether catastrophic forgetting indeed occurs. To address this, similarly to the previous point, it would be beneficial to include performance metrics before, after, and at intermediate stages of fine-tuning on paired data, but now with a focus on tasks where unpaired data results in better performance (if any). This would allow for a clear assessment of the trade-off between fine-tuning on paired data and knowledge retention.

If catastrophic forgetting does occur during, the authors could explore techniques to mitigate its effects. For instance, Elastic Weight Consolidation can penalize significant changes in weights critical for the original tasks, helping retain previously learned knowledge. Additionally, in progressive fine-tuning the model is gradually exposed to more paired data while intermittently retraining on unpaired data to preserve broader knowledge. These strategies would help balance the retention of general knowledge with the benefits of fine-tuning on paired data.

While the models excelled in antibody-specific tasks, they underperformed in predicting protein expression compared to general protein language models like ProtT5. The authors reason that antibody-specific models may have limitations in capturing the broader patterns necessary for accurate protein expression prediction, which are better learned by models trained on more diverse protein datasets. It would be valuable to investigate how general protein language models, such as ProtT5, perform on this benchmark when fine-tuned on antibody sequences. Indeed, while the authors demonstrate that models trained specifically on antibody sequences generally outperform general models, an important question left unexamined is how general models fine-tuned on antibody data would fare on this benchmark.

Another important related question is whether extensive fine-tuning could cause general models like ProtT5 to lose the broad evolutionary knowledge that contributes to their superior performance in certain tasks (once again, catastrophic forgetting). If this is the case, it raises the critical issue of determining the optimal balance between retaining the acquired general knowledge and fine-tuning, and how this compares to a model entirely from scratch on antibody sequences. This is particularly relevant given the high computational demands of training large language models from scratch (the authors used 32 A100 GPUs). Additionally, concerns about CO2 emissions and the importance of reducing redundant efforts further emphasize the need for optimizing the balance between training new models and fine-tuning existing ones.

Conclusion: The article presents significant advancements in the development of antibody-specific language models, with IgBert and IgT5 showing clear improvements over previous models. The work successfully demonstrates the potential of large-scale, paired antibody language models in enhancing tasks crucial to therapeutic antibody development. However, a few important questions remain unexplored, such as the potential for catastrophic forgetting during fine-tuning, how much paired data is necessary to achieve performance gains, etc. Addressing these questions could provide deeper insights into the trade-offs involved in fine-tuning antibody-specific models and further optimize their application in bioengineering contexts.

**Have the authors made all data and (if applicable) computational code underlying the findings in their manuscript fully available?**

Reviewer #1: Yes

Reviewer #2: Yes

PLOS authors have the option to publish the peer review history of their article (what does this mean?). If published, this will include your full peer review and any attached files.

Reviewer #1: **Yes: **Michael Heinzinger

Reviewer #2: **Yes: **Maria Rodriguez Martinez
---

## [Decision Letter · Decision Letter 1]

23 Sep 2024

Dear Dr Dreyer,

Thank you very much for submitting your manuscript "Large scale paired antibody language models" for consideration at PLOS Computational Biology.

As with all papers reviewed by the journal, your manuscript was reviewed by members of the editorial board and by several independent reviewers. In light of the reviews (below this email), we would like to invite the resubmission of a significantly-revised version that takes into account the reviewers' comments.

Given the comments of reviewer 2, I am keeping the request to the authors to provide a more convincing response to the remarks provided in the original submission. Please note that if this is not done properly - the paper will be rejected.

We cannot make any decision about publication until we have seen the revised manuscript and your response to the reviewers' comments. Your revised manuscript is also likely to be sent to reviewers for further evaluation.

Sincerely,

Miguel Francisco de Almeida Pereira de Rocha

Academic Editor

PLOS Computational Biology

Arne Elofsson

Section Editor

PLOS Computational Biology

Given the comments of reviewer 2, I am keeping the request to the authors to provide a more convincing response to the remarks provided in the original submission.

Reviewer's Responses to Questions

**Comments to the Authors:**

Reviewer #1: I remain a bit skeptical/surprised about the comparatively small improvement after adding paired data but also such results are interesting. From their response, I understand that the authors made the claim about the improvement from unpaired to paired training data based on sequence recovery but a) I could not find the IgT5_unpaired=92% vs IgT5=94% reconstruction accuracy in Table 1 within SOM3 (if you are referring to the "total" in the rightmost column of the supplementary table, either my PDF viewer does not render it for the table 2 in main or you forgot to include this column/total in the main text table) and b) even if I had found it, I would remain skeptical whether this improvement in reconstruction accuracy is meaningful. I think an improvement in reconstruction accuracy is a nice proxy but it only becomes really meaningful if it can be confirmed via better designs (probably the only use-case for reconstructing amino acids -but this most likely requires wet-lab testing) or if it correlates with better downstream prediction task quality (which I am not sure about after checking Table 3 in main). - All that being said: I appreciate that the authors re-phrased that they "believe" that paired data helps - I believe so, too, but apparently there remains some work to be done for fully leveraging this additional information. - Maybe make this a bit more transparent.

After cross-checking, the only thing that would be nice to add is the actual pre-training code (found only the huggingface entry with the weights and examples how to run the model but not the actual pre-training code).

The authors have addressed my other concerns from the previous revision accordingly.

Reviewer #2: I'm disappointed with the responses. While I understand that some of my comments required substantial computational resources and may be beyond the scope of the revision, I expected the authors to at least acknowledge them as directions for future work or open research questions. However, the response to my critique is reduced to just half a sentence in the Discussion. I strongly encourage the authors to improve this section by addressing these points more thoroughly.

**Have the authors made all data and (if applicable) computational code underlying the findings in their manuscript fully available?**

Reviewer #1: Yes

Reviewer #2: Yes

PLOS authors have the option to publish the peer review history of their article (what does this mean?). If published, this will include your full peer review and any attached files.

Reviewer #1: **Yes: **Michael Heinzinger

Reviewer #2: No
---

## [Editor Report · Decision Letter 2]

27 Oct 2024

PCOMPBIOL-D-24-01046R2Large scale paired antibody language modelsPLOS Computational Biology Dear Dr. Dreyer, Thank you for submitting your manuscript to PLOS Computational Biology. After careful consideration, we feel that it has merit but does not fully meet PLOS Computational Biology's publication criteria as it currently stands. Therefore, we invite you to submit a revised version of the manuscript that addresses the points raised during the review process. Please submit your revised manuscript within 30 days Dec 27 2024 11:59PM. If you will need more time than this to complete your revisions, please reply to this message or contact the journal office at ploscompbiol@plos.org. Please include the following items when submitting your revised manuscript:*
A rebuttal letter that responds to each point raised by the editor and reviewer(s). You should upload this letter as a separate file labeled 'Response to Reviewers'. This file does not need to include responses to formatting updates and technical items listed in the 'Journal Requirements' section below.*
A marked-up copy of your manuscript that highlights changes made to the original version. You should upload this as a separate file labeled 'Revised Manuscript with Track Changes'.*
An unmarked version of your revised paper without tracked changes. You should upload this as a separate file labeled 'Manuscript'. If you would like to make changes to your financial disclosure, competing interests statement, or data availability statement, please make these updates within the submission form at the time of resubmission. Guidelines for resubmitting your figure files are available below the reviewer comments at the end of this letter. We look forward to receiving your revised manuscript. Kind regards, Miguel Francisco de Almeida Pereira de RochaAcademic EditorPLOS Computational Biology Arne ElofssonSection EditorPLOS Computational Biology

Feilim Mac Gabhann

Editor-in-Chief

PLOS Computational Biology

Jason Papin

Editor-in-Chief

PLOS Computational Biology

 **Journal Requirements:** **Additional Editor Comments (if provided):****Reviewers' comments:**   **Figure resubmission:** While revising your submission, please upload your figure files to the Preflight Analysis and Conversion Engine (PACE) digital diagnostic tool, https://pacev2.apexcovantage.com/. PACE helps ensure that figures meet PLOS requirements. To use PACE, you must first register as a user. Registration is free. Then, login and navigate to the UPLOAD tab, where you will find detailed instructions on how to use the tool. If you encounter any issues or have any questions when using PACE, please email PLOS at figures@plos.org. Please note that Supporting Information files do not need this step. If there are other versions of figure files still present in your submission file inventory at resubmission, please replace them with the PACE-processed versions. **Reproducibility:** To enhance the reproducibility of your results, we recommend that authors of applicable studies deposit laboratory protocols in protocols.io, where a protocol can be assigned its own identifier (DOI) such that it can be cited independently in the future. Additionally, PLOS ONE offers an option to publish peer-reviewed clinical study protocols. Read more information on sharing protocols at https://plos.org/protocols?utm_medium=editorial-email&utm_source=authorletters&utm_campaign=protocols

---

## [Editor Report · Decision Letter 3]

18 Nov 2024

Dear Dr Dreyer,

We are pleased to inform you that your manuscript 'Large scale paired antibody language models' has been provisionally accepted for publication in PLOS Computational Biology.

Best regards,

Miguel Francisco de Almeida Pereira de Rocha

Academic Editor

PLOS Computational Biology

Arne Elofsson

Section Editor

PLOS Computational Biology

Feilim Mac Gabhann

Editor-in-Chief

PLOS Computational Biology

Jason Papin

Editor-in-Chief

PLOS Computational Biology

---

## [Editor Report · Acceptance letter]

27 Nov 2024

PCOMPBIOL-D-24-01046R3 

Large scale paired antibody language models

Dear Dr Dreyer,

I am pleased to inform you that your manuscript has been formally accepted for publication in PLOS Computational Biology. Your manuscript is now with our production department and you will be notified of the publication date in due course.

With kind regards,

Lilla Horvath
